# Preoperative CT-Based Skeletal Muscle Mass Depletion and Outcomes after Total Laryngectomy

**DOI:** 10.3390/cancers15143538

**Published:** 2023-07-08

**Authors:** Victoria Salati, Katerina Mandralis, Fabio Becce, Joachim Koerfer, Karma Lambercy, Christian Simon, François Gorostidi

**Affiliations:** 1Department of Otolaryngology, Head and Neck Surgery, Lausanne University Hospital (CHUV), University of Lausanne (UNIL), 1011 Lausanne, Switzerland; 2Department of Diagnostic and Interventional Radiology, Lausanne University Hospital (CHUV), University of Lausanne (UNIL), 1011 Lausanne, Switzerland

**Keywords:** skeletal muscle mass depletion, sarcopenia, muscle mass, muscle quality, total laryngectomy, postoperative outcomes

## Abstract

**Simple Summary:**

Sarcopenia is characterized by the loss of skeletal muscle mass and function and is common in head and neck oncology. This retrospective study aimed to quantify preoperative muscle mass and quality using CT-based indices and to assess their impact on postoperative outcomes and survival in patients who underwent total laryngectomy for cancer. Based on pre-established cut-off values, 44% of the patients in this series had preoperative skeletal muscle mass depletion. No association was found between CT-based skeletal muscle mass depletion alone and postoperative outcomes or overall survival after total laryngectomy.

**Abstract:**

Purpose: To assess the role of preoperative CT-based skeletal muscle mass depletion on postoperative clinical outcomes and survival in patients who underwent total laryngectomy for cancer. Methods: Patients operated on between January 2011 and March 2020 were retrospectively included. Skeletal muscle area and intra- and inter-muscular fat accumulation were measured at the third lumbar vertebral level on preoperative CT scans. Skeletal muscle mass depletion was defined based on pre-established cut-off values. Their association with postoperative morbidity, length of stay (LOS), costs, and survival was assessed. Results: A total of 84 patients were included, of which 37 (44%) had preoperative skeletal muscle mass depletion. The rate of postoperative fistula (23% vs. 35%, *p* = 0.348), cutaneous cervical dehiscence (17% vs. 11%, *p* = 0.629), superficial incisional surgical site infections (SSI) (12% vs. 10%, *p* = 1.000), and unplanned reoperation (38% vs. 37%, *p* = 1.000) were comparable between the two patient groups. No difference in median LOS was observed (41 vs. 33 days, *p* = 0.295), nor in treatment costs (119,976 vs. 109,402 CHF, *p* = 0.585). The median overall survival was comparable between the two groups (3.43 vs. 4.95 years, *p* = 0.09). Conclusions: Skeletal muscle mass depletion alone had no significant impact on postoperative clinical outcomes or survival.

## 1. Introduction

Due to tumor location, head and neck cancer (HNC) patients are extremely prone to weight loss prior to diagnosis, during treatment, and after treatment completion [1]. Malnutrition, tumor burden, immobilization, and treatment-related adverse events are all risk factors for developing cancer cachexia and sarcopenia. According to the European Working Group on Sarcopenia in Older People (EWGSOP), sarcopenia is defined as the generalized loss of skeletal muscle mass and quality and is considered severe when low physical performance is detected [2].

Computed tomography (CT) at the third lumbar (L3) vertebral level is considered one of the reference standards for non-invasive measurement of body composition in cancer patients [2]. The skeletal muscle area (SMA) and index (SMI) at L3 are validated muscle quantity indices and have been shown to correlate strongly with whole-body muscle mass [3]. Intramuscular fat accumulation, or myosteatosis, and intermuscular but extracellular (subfascial) lipid storages can be assessed by the decrease in skeletal muscle radiation attenuation (SMRA) in the Hounsfield unit (HU) and increase in intermuscular adipose tissue index (IMATI), respectively [4,5].

Sarcopenia has been recognized as a poor prognostic factor for postoperative complications, rate of chemoradiation completion, and survival in HNC patients [6,7,8,9,10,11,12]. Specifically after total laryngectomy for advanced laryngeal carcinoma, sarcopenia has been shown to predict postoperative wound complications and pharyngo-cutaneous fistulas (PCF) [10,13,14,15].

Currently, while a wide variety of diagnostic methods have been used to assess body composition, mainly muscle mass indices were evaluated, and only few studies have investigated the effects of muscle quality indices (SMRA and IMATI) [16,17,18,19]. The correlation between muscle mass and quality is not well understood in cancer patients and is an area of active research. Moreover, a single study has yet evaluated the impact of sarcopenia on surgical treatment costs [20].

The objectives of this study were to quantify the preoperative muscle quantity (SMI) and quality (SMRA and IMATI) in HNC patients who underwent total laryngectomy for cancer and to assess the impact of preoperative CT-based skeletal muscle mass (SMM) depletion on postoperative clinical outcomes and survival.

## 2. Materials and Methods

### 2.1. Study Design and Patients

This single-center retrospective cohort study included all consecutive patients who underwent total laryngectomy for cancer between January 2011 and March 2020 in the Department of Otolaryngology and Head and Neck Surgery of Lausanne University Hospital (CHUV), Lausanne, Switzerland. All treatment decisions were discussed at a weekly multidisciplinary board meeting. Patients included were those who underwent total laryngectomy as primary treatment for cancer or as salvage surgery in the event of previous (radio)chemotherapy or partial surgery with cancer recurrence. Patients were included if a preoperative abdominal CT or PET/CT scan was available in the 3 months prior to surgery. Patients who underwent partial laryngectomies for cancer and patients who underwent total laryngectomies for dysfunctional larynx without cancer were excluded.

The retrieved patients’ demographics were age, sex, body mass index (BMI), smoking status, active alcohol consumption, comorbidities, American Society of Anesthesiologist (ASA) scores, and treatment history. Tumor subsite, 7th and 8th TNM staging according to the American Joint Committee on Cancer (AJCC), the association of partial pharyngectomy, neck dissection, and the use of a flap for reconstruction were collected.

### 2.2. CT-Based Muscle Quantity and Quality

Skeletal muscle quantity and quality indices were calculated from the psoas, paraspinal, and abdominal wall muscles using single axial CT images of the abdomen at L3 level (Figure 1). Initial muscle segmentations were automatically generated using a deep-learning-based algorithm [21], which was developed and trained for abdominal muscle segmentation on L3 CT images from cancer patient populations. All original muscle segmentations were then reviewed and corrected manually as necessary by two trained radiologists using a custom graphical user interface in the same fashion as in previous studies [4,22,23]. The cross-sectional area of the aforementioned muscles (only pixels in the range of −29 to +150 HU) represented the SMA (cm^2^) and was then normalized by patient height squared to obtain the SMI (cm^2^/m^2^). Pre-established, sex-specific SMI cut-off values (females < 38.5 cm^2^/m^2^, males < 52.4 cm^2^/m^2^) were used to determine whether sarcopenia was present or not, as previously reported [24,25]. Two muscle quality indices were further calculated from the corrected muscle segmentations, the SMRA in HU and the intermuscular adipose tissue (IMAT) in cm^2^. The IMAT was also normalized by patient height squared to obtain the IMATI (cm^2^/m^2^) [26].

### 2.3. Outcomes

Postoperative clinical outcomes included length of stay (LOS), number of unplanned reoperations, treatment costs (including hospitalization) in Swiss francs (CHF), and number of wound-related complications occurring within 30 days from surgery: pharyngo-cutaneous fistula (PCF), superficial incisional surgical site infections (SSI), and cutaneous cervical dehiscence. PCF was defined as a clinical salivary leak. In our institution, all fistulas are managed surgically. A cumulative variable grouping of all wound-related complications was created, including fistulas, cutaneous dehiscence, and superficial incisional SSI. Unplanned reoperations consisted in reopening the wounds with exploration of the surgical site. Follow-up and survival data were collected until June 2022. Five-year overall survival was calculated from the date of total laryngectomy to the date of death or loss of follow-up.

Patients with vs. those without SMM depletion were compared in terms of demographics, clinical outcomes, and survival.

### 2.4. Statistical Analysis

Statistical analyses were performed using SPSS 26 (SPSS Inc., Chicago, IL, USA) and RStudio (version 2022.12.0+353). Categorical variables were expressed as number and frequencies (%) and compared between patient groups with Pearson’s chi-squared or Fisher’s exact test, where appropriate. Continuous variables were expressed as mean (standard deviation, SD) or median (interquartile range, IQR) and compared with the Mann–Whitney U test or Student’s *t*-test according to their distribution. Receiver operating characteristic (ROC) curves were used to assess the performance of skeletal muscle mass and quality depletion indices as a diagnostic test to predict the occurrence of postoperative pharyngo-cutaneous fistulas and unplanned reoperations. A good diagnostic performance of a test was defined as an ROC curve having an area under the curve ≥ 0.7. Logistic binary regression was used for predictive factors of pharyngo-cutaneous fistulas. Overall survival was analyzed with the Kaplan–Meier method, and groups were compared with the log-rank test. Statistical significance was set at a *p*-value < 0.05.

## 3. Results

Among the 84 patients included, 37 (44%) had preoperative CT-based SMM depletion. Patient demographics, treatment history, and surgical details are described in Table 1. Patients with SMM depletion had lower mean BMI (20.2 kg/m^2^ vs. 24.3 kg/m^2^, *p* < 0.003), lower mean SMA (119.3 cm^2^ vs. 146.2 cm^2^, *p* < 0.001), lower mean IMAT (13.7 cm^2^/m^2^ vs. 19.4 cm^2^/m^2^, *p* = 0.029), and lower mean IMATI (4.4 vs. 6.7 cm^2^, *p* = 0.007) compared to the group without SMM depletion. A total of 21% (n = 18) of the cohort was underweight according to BMI (<18.5 kg/m^2^). There was no difference in SMRA (41.2 HU vs. 39.7 HU, *p* = 0.360) between the two patient groups. Patients with and without SMM depletion were comparable in terms of comorbidities, treatment history, tumor location, and surgical details.

Postoperative clinical outcomes are described in Table 2. The two patient groups were comparable in terms of median LOS (28 days vs. 28 days, *p* = 0.901), cutaneous dehiscence (8% vs. 4%, *p* = 0.629), superficial incisional SSI (6% vs. 4%, *p* = 1.000), unplanned reoperations (18% vs. 14%, *p* = 1.000), and mean treatment costs (119,976 vs. 109,402 CHF, *p* = 0.585). Regarding the occurrence of postoperative fistula, there was no significant difference between patients with or without SMM depletion (11 vs. 13%, *p* = 0.348). None of the SMM depletion and fat accumulation indices (SMI, SMRA, IMATI) had a clinically meaningful diagnostic performance to predict the occurrence of postoperative fistulae, all wound-related complications, and unplanned reoperations according to the ROC curve analyses (Figure 2). On univariate analyses, independent risk factors for postoperative fistula were prior locoregional radiotherapy (OR 3.25, *p* = 0.019), prior chemotherapy (OR 3.77, *p* = 0.012), flap reconstructions (OR 2.60, *p* = 0.037), and salvage surgeries (OR 3.89, *p* = 0.007). However, none remained statistically significant on multivariate analyses (Table 3). Univariate and multivariate analyses of factors associated with the cumulative variable of all wound-related complications can be found in Appendix A. On multivariate analyses, flap reconstruction remained statistically significant as an independent risk factor for all wound-related complications (OR 2.91, *p* = 0.027).

The difference in survival between patients with vs. those without preoperative SMM depletion was not significant with the log-rank test (*p* = 0.09), with a median survival of 3.43 years (95% CI 0.89–5.96) in the group with SMM depletion compared to 4.95 years (95% CI 4.08–5.82) for the group without SMM depletion (Figure 3).

## 4. Discussion

This study showed that 44% of HNC patients who underwent total laryngectomy had preoperative SMM depletion, regardless of their treatment history and unrelated to their major comorbidities. Preoperative SMM depletion alone was not associated with decreased overall survival after laryngectomy or with other adverse postoperative clinical outcomes. Further research, considering not only CT-based SMM and quality indices but also muscle strength and physical performance, is thus needed to better stratify the preoperative risk in HNC patients.

The high incidence of sarcopenia in our cohort corroborates the high prevalence of malnutrition in HNC patients among all cancer patient populations [27]. However, weight loss alone usually measured by the BMI is recognized as an inaccurate marker of nutritional status and survival prognosis [28]. Indeed, normal-weight and obese patients may have occult sarcopenia due to disparate accumulation and loss of fat and muscle tissue, respectively [25]. In our study, although patients with sarcopenia had significantly lower BMI, only 21% of the cohort were underweight according to BMI whereas 44% had SMM depletion. In a retrospective study including 235 patients undergoing total laryngectomy, BMI-specific cut-off values for low skeletal muscle mass could only discriminate 80% of sarcopenic patients as defined by CT-based body composition markers [10]. A review of 1473 upper gastrointestinal and lung cancer patients showed that marked weight loss, low SMA, and low SMRA were independent prognostic factors for poorer survival regardless of BMI [28].

Among the various tools available for in vivo assessment of body composition, CT has been increasingly used in research and clinical settings, probably due to its wide availability in oncology patients. In our study, a deep learning model provided original muscle segmentations from a single axial CT image at L3. This standardized method was previously trained on CT images extracted at the L3 level and has been proven to be relatively accurate and reliable across different cancer patient populations [4,23]. Some authors advocate the use of cross-sectional CT images obtained at the third cervical (C3) vertebral level for the HNC population. Swartz et al. found a strong correlation between paravertebral and sternocleidomastoid muscle SMA at the C3 level and SMA at the L3 level [29]. However, C3 to L3 conversion formulas can be complex to use in practice, and several studies have directly evaluated outcomes with measurements at C3 [14,15,30]. Limitations of CT images at C3 include dental artefacts and exclusion of part of the sternocleidomastoid muscles or manual approximation of their segmentation due to tumor invasion [14]. However, for increased reproducibility, the use of the C3 level may be favored over L3 in the HNC population because some patients do not undergo abdominal CT scans as part of their initial imaging work-up. Along with the interest in including a maximum number of patients, a combination of cervical magnetic resonance imaging (MRI) and CT has been used to estimate the SMA [10,31]. While a strong correlation between CT- and MRI-derived SMA values has been obtained mainly in breast cancer research, no comparative study between the two modalities has been performed so far in the HNC population [32]. MRI, with its higher soft tissue contrast resolution, could provide more data regarding muscle quality parameters such as differentiation between intra- and inter-myocellular fat, edema, and scars. However, such advanced MRI protocols are currently not yet used in routine clinical practice [32].

There is great variability between studies regarding the diagnostic cut-off values for low skeletal muscle mass, mainly because they were obtained based on different patient populations (cancer or healthy, sex-specific or not) and different dependent variables [2,10,33,34]. Indeed, cut-off values can be selected based on the population or outcomes, using SMI ROC curves or tercile thresholds [15,35,36]. In the present study, the cut-off values used (women <38.5 cm^2^/m^2^, men <52.4 cm^2^/m^2^) were determined by optimal stratification in previously published studies of cancer populations at the L3 level [25]. To date, there is still no evidence-based consensus on the lower limit of the SMI as a diagnostic criterion for sarcopenia [37]. Reference values also vary by sex, ethnicity, age, tumor types, and outcomes, thereby limiting comparisons [2,33,38]. The HU range for the measurement of SMRA also lacks diagnostic thresholds, further hindering inter-group comparison [26].

In the present cohort, major comorbidities, treatment history, and tumor location did not differ between the two patient groups (SMM depletion or not). The incidence of sarcopenia between different stages of cancer could not be compared by subgroup analyses, because the patients laryngectomized for malignancies were mainly in advanced stages.

Surprisingly, the two calculated muscle quality parameters, SMRA and IMATI, were not in agreement with muscle quantity indices. In the present study, sarcopenic patients, as defined by SMI, did not have significantly lower SMRA but had significantly lower IMATI (Table 1). The lack of correlation between muscle quantity and quality parameters may support the notion that SMM depletion and myosteatosis may represent two distinct disease processes or phenotypes and different patient groups [39]. In a retrospective review of 225 HNC patients undergoing (radio)chemotherapy, low SMRA and high IMATI were significantly associated with poorer overall survival, progression-free survival, and cause-specific survival in a multivariate analysis, whereas SMI was not a prognostic factor [40]. Lower IMAT/IMATI in our cohort may be related to their lower BMI and underlying cancer cachexia. However, decreased SMRA has been described as a feature of advanced cancers and associated with poorer muscle function and shorter overall survival in various cancer populations [16,19,41]. Shaver et al. linked myosteatosis to decreased quality of life in HNC patients one year after treatment completion [42].

Several studies investigating post-laryngectomy outcomes have found that sarcopenia was useful in predicting wound complications and pharyngo-cutaneous fistulas [10,13,14,15]. The present study did not confirm these results, as there was no significant difference between patient groups in terms of wound complications (cutaneous dehiscence, superficial incisional SSIs, pharyngo-cutaneous fistulas, and unplanned reoperations) nor in terms of LOS and costs. Pharyngo-cutaneous fistula has been reported as the most common complication after total laryngectomy, reaching incidence rates of 21% for laryngectomy as primary treatment and 29% for salvage surgery in two meta-analyses including 8605 and 3292 patients, respectively [43,44]. In the present cohort composed of 60% of primary laryngectomies and 40% of salvage surgery, the postoperative rate of pharyngo-cutaneous fistulas was 28%. The overall incidence of superficial incisional SSIs (12%) and cutaneous dehiscence (14%) was consistent with the incidence reported in meta-analyses [44]. A retrospective cohort study including 400 total, supraglottic, and partial laryngectomies reported 13% of unplanned reoperations, but their highest reoperation rate was in the pharyngectomy group (17%) [45]. The higher total unplanned reoperation rate (39%) in our study compared with the reported rates might be related to the fact that pharyngo-cutaneous fistulas are managed surgically according to the local protocol. None of the sarcopenia indices were able to predict the risk of fistula occurrence in the ROC curve analyses.

By reviewing the literature, there are arguments for decreased overall survival in sarcopenic HNC patients based on SMM depletion measures [35,46,47,48]. Two meta-analyses, each including more than 2000 patients, confirmed the negative impact of preoperative sarcopenia on overall survival [12,46]. The exact mechanism underlying these results is not yet fully understood. However, lower tolerance to treatments, higher susceptibility to complications, and worse quality of life may be contributing factors. Our study reports contradictory results, with no decreased overall survival at 5 years after total laryngectomy for patients with SMM depletion based on CT-alone.

Beyond its retrospective design, several limitations must be acknowledged in the present study. First, the median time from CT image acquisition to total laryngectomy was 37 days, potentially not being representative of the patients’ status at the time of surgery. Second, there may be variability in skeletal muscle densities (i.e., SMRA) between non-contrast and contrast-enhanced CT scans for density measurements. In our cohort, both non-contrast and contrast-enhanced CT and PET/CT scans were used, potentially contributing to data heterogeneity and clinically relevant differences. Patients who underwent total laryngectomy for laryngeal dysfunction were included, which also contributes to data heterogeneity in the study population. Third, muscle mass and quality indices should be evaluated as a dynamic process over time, with reassessment after systemic treatments, radiotherapy, and surgery, and could be compared to a control group. Indeed, Stene et al. showed that an increase in muscle mass during palliative chemotherapy for advanced lung cancer was associated with longer survival but not baseline muscle mass [49]. In 125 HNC patients undergoing curative-intent treatment, post-treatment sarcopenia was associated with poorer 5-year overall survival [50]. However, our study focused on preoperative sarcopenia, without further analysis and comparison over time. Finally, sarcopenia was only assessed by CT data, and no functional evaluation was performed, although the latter is part of the full definition of sarcopenia according to the revised EWGSOP guidelines [2].

## 5. Conclusions

The present study showed no significant impact of preoperative CT-based SMM depletion alone on postoperative clinical outcomes or overall survival after total laryngectomy. CT-based muscle mass and quality assessment alone, without taking into account muscle strength and physical performance, should thus be considered with caution in clinical practice when stratifying preoperative risk and adapting postoperative interventions to improve oncological outcomes.

## Figures and Tables

**Figure 1 cancers-15-03538-f001:**
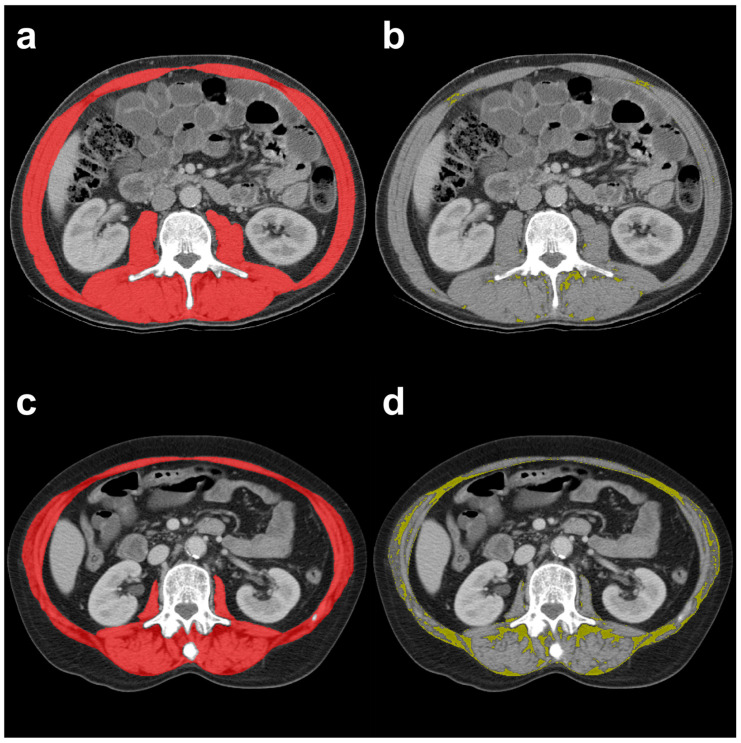
Representative examples of CT images at L3 level showing segmentation of the SMA ((**a**,**c**), color-coded in red) with secondary derivation of the IMAT aera ((**b**,**d**), color-coded in yellow) in two HNC patients who underwent total laryngectomy with favorable (top row) and unfavorable (bottom row) postoperative clinical outcomes.

**Figure 2 cancers-15-03538-f002:**
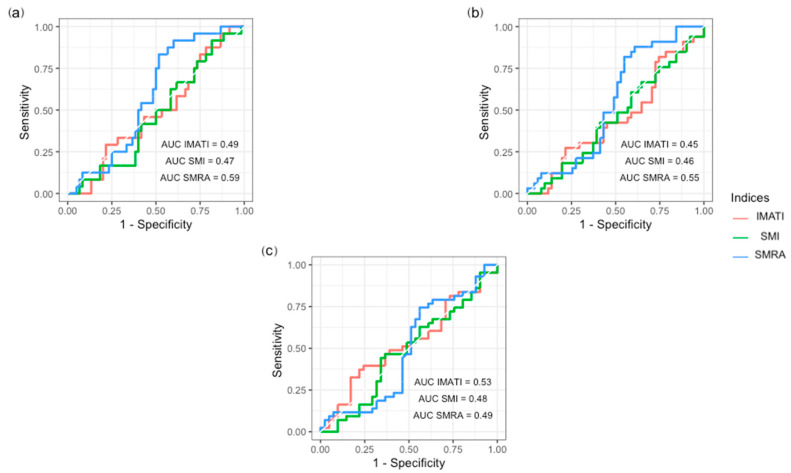
Receiver operating characteristic (ROC) curves of preoperative skeletal muscle mass and quality indices for predicting postoperative fistula, all wound-related complications, and unplanned reoperations: (**a**) postoperative fistula; (**b**) all wound-related complications; (**c**) unplanned reoperations.

**Figure 3 cancers-15-03538-f003:**
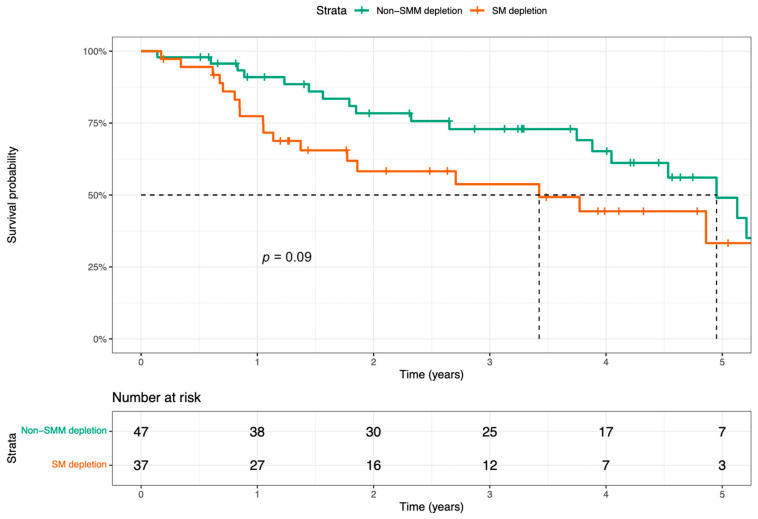
Kaplan–Meier curves of overall survival after total laryngectomy in HNC patients with vs. those without preoperative SMM depletion.

**Table 1 cancers-15-03538-t001:** Preoperative patient characteristics, prior treatment, and surgical details.

	Non-SMM Depletion (n = 47)	SMM Depletion (n = 37)	*p*-Value *
Age (years), (mean, SD)	65 (9)	64 (9)	0.425
Sex (male), (n, %)	36 (76.5)	32 (86.5)	0.386
BMI (kg/m^2^), (mean, SD)	24.3 (6.8)	20.2 (4.8)	**<0.003**
SMA (cm^2^), (mean, SD)	146.2 (29.4)	119.3 (23.0)	**<0.001**
SMRA (HU), (mean, SD)	39.7 (8.0)	41.2 (6.8)	0.360
IMAT (cm^2^), (mean, SD)	19.4 (12.8)	13.7 (10.0)	**0.029**
IMATI (cm^2^/m^2^), (mean, SD)	6.7 (4.3)	4.4 (3.1)	**0.007**
Cardiovascular disease, (n, %)	13 (27.7)	7 (18.9)	0.499
Pulmonary disease, (n, %)	16 (34.0)	16 (43.2)	0.525
Renal disease, (n, %)	5 (10.6)	3 (8.1)	0.986
Diabetes, (n, %)	6 (12.8)	3 (8.1)	0.741
ASA score ≥ 3, (n, %)	32 (68.1)	24 (64.9)	0.938
Smoking, (n, %)	20 (46.5)	20 (57.1)	0.480
Alcohol consumption, (n, %)	25 (53.2)	28 (75.7)	0.089
Prior locoregional RT, (n, %)	17 (36.1)	18 (48.6)	0.396
Prior chemotherapy, (n, %)	11 (23.4)	11 (29.7)	0.686
Surgical indication, (n, %)			0.664
Primary	30 (63.8)	21 (56.8)	
Salvage	17 (36.2)	16 (43.2)	
Tumor location, (n, %)			0.630
Laryngeal	31 (66.0)	24 (64.9)	
Pharyngeal	9 (19.1)	5 (13.5)	
Laryngo-pharyngeal	7 (14.9)	8 (21.6)	
Procedure, (n, %)			0.660
Total laryngectomy	25 (53.2)	15 (45.9)	
Pharyngolaryngectomy	22 (46.8)	20 (54.1)	
Neck dissection, (n, %)	44 (93.6)	35 (94.6)	1.000
Flap reconstruction, (n, %)	21 (44.7)	19 (51.4)	0.698

* Significant *p*-values (<0.05) are displayed in bold characters. SD: standard deviation; SMM: skeletal muscle mass; BMI: body mass index; SMA: skeletal muscle area; SMRA: skeletal muscle radiation attenuation; HU: Hounsfield unit; IMAT: intermuscular adipose tissue; IMATI: intermuscular adipose tissue index; ASA: American Society of Anesthesiologists; RT: radiotherapy.

**Table 2 cancers-15-03538-t002:** Postoperative clinical outcomes.

	Non-SMM Depletion (n = 47)	SMM Depletion (n = 37)	*p*-Value *
Length of stay (days), (median, IQR)	28 (19–45)	28 (21–37)	0.901
Fistula, (n, %)	11 (23.4)	13 (35.1)	0.348
Cutaneous dehiscence, (n, %)	8 (17.0)	4 (10.8)	0.629
Superficial incisional SSI, (n, %)	6 (12.8)	4 (10.8)	1.000
Unplanned reoperation, (n, %)	18 (38.3)	15 (40.5)	1.000
Fistula	11 (61.1)	13 (86.7)	0.294
Cutaneous dehiscence	2 (11.1)	2 (13.3)	
Chyle leak	3 (16.7)	0	
Hematoma	2 (11.1)	0	
Treatment costs (CHF), (mean, SD)	119,976 (102,454)	109,402 (64,107)	0.585

* Significant *p*-values (<0.05) are displayed in bold characters. IQR: interquartile range; SSI: surgical site infections; CHF: Swiss francs.

**Table 3 cancers-15-03538-t003:** Univariate and multivariate analyses of factors associated with fistulas.

	Univariate		Multivariate	
	OR (95% CI)	*p*-Value *	OR (95% CI)	*p*-Value *
Age	0.97 (0.92–1.02)	0.244		
Male sex	0.85 (0.27–3.00)	0.792		
BMI	0.98 (0.91–1.06)	0.580		
ASA ≥3	1.31 (0.48–3.84)	0.609		
Smoking	1.20 (0.45–3.28)	0.718		
SMM depletion	1.77 (0.68–4.68)	0.240		
SMI	0.88 (0.23–2.80)	0.880		
SMRA	1.04 (0.98–1.12)	0.219		
IMATI	0.96 (0.84–1.08)	0.521		
Prior neck dissection	0.45 (0.06–1.87)	0.322		
Prior locoregional RT	3.25 (1.23–9.01)	**0.019**	0.67 (0.02–7.55)	0.766
Prior chemotherapy	3.77 (1.34–10.84)	**0.012**	1.31 (0.26–7.09)	0.749
Prior tracheostomy	2.15 (0.68–6.63)	0.184		
Flap reconstruction	3.00 (1.14–8.45)	**0.030**	1.87 (0.56–6.19)	0.301
Procedure				
Total laryngectomy	-	-		
Pharyngolaryngectomy	1.60 (0.62–4.26)	0.336		
Neck dissection	1.64 (0.23–33.06)	0.665		
Tumor location				
Laryngeal	-	-		
Pharyngeal	1.79 (0.48–6.22)	0.362		
Laryngo-pharyngeal	2.15 (0.62–7.18)	0.212		
Surgical indication				
Primary	-	-		
Salvage	3.89 (1.14–10.87)	**0.007**	3.64 (0.28–95.5)	0.350

* Significant *p*-values (<0.05) are displayed in bold characters.

## Data Availability

No new data were created or analyzed in this study. Data sharing is not applicable to this article.

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
