# Peer review of "Preoperative CT-Based Skeletal Muscle Mass Depletion and Outcomes after Total Laryngectomy"

_cancers, 2023, doi:10.3390/cancers15143538_

Round 1

Reviewer 1 Report

This paper shows lower survival in patients with CT-defined sarcopenia in HNC patients who undergo total laryngectomy. No impact of sarcopenia on postoperative fistula was seen. The result is interesting, although the research question is not particularly novel. These results add to the growing literature on the adverse impact of sarcopenia on cancer outcomes.

However, some issues need clarification.

1. The rate of malignancy was 92% in the non-sarcopenic group and 76% in the sarcopenic group (Table 1). In the Discussion section, it is stated that "the rate of malignancy was significantly higher in patients with sarcopenia..." which is contradictory to the data in Table 1. Furthermore, all patients are supposed to be HNC patients according to the inclusion criteria, therefore, all should have "malignancy". Finally, in the Discussion section, it is mentioned that "...patients laryngectomized for malignancies...", although all patients should have been treated for malignancy. This issue requires a thorough clarification with regard to inclusion criteria and what is meant by "malignancy" here.

2. The generalizability of the results is limited by the fact that most HNC patients undergo imaging of the neck and thorax only, not the lumbar region. Although the authors discuss the limitations of comparing C3 and L3 measurements, the actual rationale of using L3 and not C3 in these patients is not really mentioned. This issue needs to be better discussed.

Author Response

  1. The rate of malignancy was 92% in the non-sarcopenic group and 76% in the sarcopenic group (Table 1). In the Discussion section, it is stated that "the rate of malignancy was significantly higher in patients with sarcopenia..." which is contradictory to the data in Table 1. Furthermore, all patients are supposed to be HNC patients according to the inclusion criteria, therefore, all should have "malignancy". Finally, in the Discussion section, it is mentioned that "...patients laryngectomized for malignancies...", although all patients should have been treated for malignancy. This issue requires a thorough clarification with regard to inclusion criteria and what is meant by "malignancy" here.

Thank you for the comment. In the revised manuscript, the inclusion criteria were modified to only include patient treated for cancer. The patients who underwent total laryngectomy for dysfunctional larynx without active cancer were excluded. The manuscript has been corrected accordingly.

  1. The generalizability of the results is limited by the fact that most HNC patients undergo imaging of the neck and thorax only, not the lumbar region. Although the authors discuss the limitations of comparing C3 and L3 measurements, the actual rationale of using L3 and not C3 in these patients is not really mentioned. This issue needs to be better discussed.

Thank you for the comment. In this study, the rationale of using L3 and instead of C3 was the use of a a deep-learning-based algorithmfor initial muscle segmentation on scans, which were then reviewed by radiologists. This algorithm was trained on L3 on previous cancer populations. In the revised manuscript, this point has been better clarified in the Materials and Methods and Discussion sections.

Reviewer 2 Report

Sarcopenia has been recognized as a poor prognostic factor for postoperative complications, rate of chemoradiation completion, and survival in HNC patients. Specifically after total laryngectomy for advanced laryngeal carcinoma, sarcopenia has been shown to predict postoperative wound complications and pharyngo-cutaneous fistulas.

The objectives of this study were to quantify the preoperative muscle quantity and quality (SMRA and IMATI) in HNC patients who underwent total laryngectomy and to assess the impact of preoperative CT-based sarcopenia on postoperative clinical out-comes and survival.

The study is very interesting, well written methodologically.

This study showed that half of the patients who underwent total laryngectomy had preoperative sarcopenia, regardless of their treatment history and unrelated to their major comorbidities. Preoperative sarcopenia was associated with decreased overall survival after laryngectomy but had no impact on other postoperative clinical outcomes.

The high incidence of sarcopenia in cohort corroborates the high prevalence of malnutrition in HNC patients among all cancer patient populations. Very interesting is that weight loss alone usually measured by the BMI is recognized as an inaccurate marker of nutritional status and survival prognosis.

Reviewing the literature and in agreement with the findings of study, there is a strong argument for decreased overall survival in sarcopenic HNC patients.

However, the evaluation made three months before surgery is not indicative and presents problems because it varies over time.

The execution of the preoperative abdominal CT requires an additional examination often not required in the staging of head and neck cancer with an increase in health care costs. It would therefore be desirable, as proposed by other AA, the use of cross-sectional CT images obtained at the third cervical (C3) vertebral level for the HNC population.

A control group composed of normal subjects should be used to evaluate whether there are variations in muscle quantity (SMI) and quality (SMRA and IMATI) over the months.

Minor editing of English language required.

Author Response

Thank you for the comment. In the revised manuscript, the inclusion criteria have been modified to only include patients who underwent total laryngectomy for cancer. By excluding the group who did not have cancer (“dysfunctional larynx”), the median time interval between CT and surgery was of 37 days.

In this study, patients were included if an abdominal CT scan was available as the deep-learning-based algorithm for initial muscle segmentation on scans was trained on L3 levels in cancer populations. This point has been better explained in the Materials and Methods and in the Discussion sections. 

The use of cross-sectional CT images obtained at the third cervical (C3) vertebral level is an interesting alternative but has not been used in this study. The first reason was the use of the deep-learning-based algorithm for initial muscle segmentation which was trained on L3 levels in cancer populations and not trained on other vertebral levels. Then, for reproducibility, as the cut-off we used were established in previously published studies of cancer population at L3. Finally, the various limitations of the use of C3 as dental artefacts, direct muscle invasion by tumor nodes have been explained in the Discussion section.

Thank you for your comment regarding variations in muscle quantity (SMI) and quality (SMRA and IMATI) over time and comparison to a control group. In this study, we focused on the baseline timepoint at the preoperative imaging workup with no further comparison over time or against control groups. A study focusing on the evolution of body composing after oncological treatments and compared to non-cancer patients is an area of further research. This point has been added to the Discussion section.

Reviewer 3 Report

This a large, monocentric, retrospective analysis on a cohort of patients submitted to total laryngectomy (TL) on a 9-year period (between January 2011 and March 2020) with specific focus on the effect of depletion/change in skeletal muscle mass (SMM) as surrogate marker of sarcopenia on survival and specific postoperative clinical outcomes, such as length of hospital stay (LOS), number of unplanned reoperations, treatment costs (CHF), number of wound-related complications (pharyngocutaneous fistula or PCF, superficial incisional surgical site infection or SSI, cutaneous cervical dehiscence). Sarcopenia was assessed on cross-sectional imaging (CT, PET/CT) at L3 level either through skeletal muscle index (SMI, cm2/m2), skeletal muscle radiation attenuation (SMRA, HU) and intermuscular adipose tissue index (IMATI, cm2/m2). Pre-established sex-specific cut-offs were used to determine whether sarcopenia was present or not (<38.5 cm2/m2 for females, <52.4 cm2/m2 for males).

The authors demonstrated that patients with preoperative sarcopenia had a decreased overall survival after TL, whereas postoperative outcome on a short-term basis was not impaired. I do appreciate the authors’ effort to try to address a major concern of modern head and neck oncology, which is the role of frailty in affecting the outcome of major surgical procedures. However, some points are critical:

- The authors analyzed the depletion of SMM, not sarcopenia itself (which should also include a functional evaluation); I think this point should be clearly stated throughout the manuscript (and in the title), not only in the final part of the Discussion.

- In M&M section the authors stated that PCF was defined as “a clinical salivary leak requiring any form of reoperation or conservative treatment”. Conversely, in the Discussion the authors stated that PCF “are managed surgically according to the local protocol”. How are these sentences coherent?

- I do think the authors should also analyze major PCF per se, meaning only those that made necessary a reoperation, since these ones are the real problem in terms of management. Perhaps they would have achieved different results as far as depletion of SMM was concerned.

- To avoid bias, I would suggest the authors to exclude patients who underwent TL for dysfunctional larynx, since these are very different subjects from a clinical standpoint and, possibly, also in terms of SMM

- The authors assessed sarcopenia through pre-established sex-specific cut-offs (<38.5 cm2/m2 for females, <52.4 cm2/m2 for males); I would suggest them trying to establish their own thresholds with ROC curves (for instance, taking advantage of “major complication” – Clavien-Dindo ≥ 3 – as outcome) and maybe also to evaluate the depletion of SMM as a continuous variable.

- Unplanned reoperation should be detailed according to causes and surgical procedures

- A cumulative variable with all the wound-related complications together (PCF, SSI, cutaneous cervical dehiscence as well as flap failure and reoperations) should be created to better assess the role of SMM depletion as a risk factor

- As declared, the median time from CT image acquisition to total laryngectomy varied up to three months before surgery, potentially not being representative of the patients’ status at the time of surgery: this could have jeopardized the role of SMM depletion as a risk factor. A more limited inclusion criterion could be used.

- A figure with an example of muscular segmentation and definition of skeletal muscle area (SMA), SMRA and intermuscular adipose tissue (IMAT) should be added for the sake of comprehension

After all these considerations, I do think this is a work that should be considered for publication after having revised the mentioned points.

Author Response

- The authors analyzed the depletion of SMM, not sarcopenia itself (which should also include a functional evaluation); I think this point should be clearly stated throughout the manuscript (and in the title), not only in the final part of the Discussion.

Thank you for the comment. As this study did not evaluate muscle strength or low physical performance, the term “CT-based skeletal muscle mass depletion” has been used rather than sarcopenia, to be in accordance with expert’s consensus about the definition of sarcopenia.

- In M&M section the authors stated that PCF was defined as “a clinical salivary leak requiring any form of reoperation or conservative treatment”. Conversely, in the Discussion the authors stated that PCF “are managed surgically according to the local protocol”. How are these sentences coherent?

Thank you for highlighting this confusion. As this was a retrospective study over 9 years, we did not know at inclusion if all fistulas had been managed surgically over the years. However, it appears that all patients with a fistula underwent surgical treatment. The Materials and Methods section has been modified accordingly.

- I do think the authors should also analyze major PCF per se, meaning only those that made necessary a reoperation, since these ones are the real problem in terms of management. Perhaps they would have achieved different results as far as depletion of SMM was concerned.

Given the local habits of treating all fistula surgically, we were not able to stratify pharyngocutaneous fistula regarding their management.

- To avoid bias, I would suggest the authors to exclude patients who underwent TL for dysfunctional larynx, since these are very different subjects from a clinical standpoint and, possibly, also in terms of SMM

In the revised manuscript, the inclusion criteria have been modified to only include patient treated for cancer. The patients who underwent total laryngectomy for dysfunctional larynx without cancer have been excluded. The manuscript has been corrected accordingly.

- The authors assessed sarcopenia through pre-established sex-specific cut-offs (<38.5 cm2/m2for females, <52.4 cm2/m2 for males); I would suggest them trying to establish their own thresholds with ROC curves (for instance, taking advantage of “major complication” – Clavien-Dindo ≥ 3 – as outcome) and maybe also to evaluate the depletion of SMM as a continuous variable.

Thank you for this interesting comment. The three skeletal muscle mass depletion and quality indices (SMI, SMRA, IMATI) have been tested using Receiver Operating Characteristics curves for predicting the occurrence of fistula, all wound related complications, and unplanned reoperations. None of the indices had a clinically meaningful diagnostic performance (AUC ≥ 0.7) in predicting any of these outcomes to establish our own thresholds.

The Clavien-Dindo grading of complications was not available in patients’ records and we did not have enough information to reliably calculate them retrospectively.

The SMI, SMRA and IMATI as continuous variables were compared to post-operative outcomes in univariate and multivariate analysis against fistula occurrence (Table 3.) and all wound-related complications (supplementary material Table 4).

- Unplanned reoperation should be detailed according to causes and surgical procedures

All unplanned reoperation consisted in reopening the wounds and exploring the surgical site. The Materials and Methods sections was revised accordingly. Causes of unplanned reoperations were added to Table 2.

- A cumulative variable with all the wound-related complications together (PCF, SSI, cutaneous cervical dehiscence as well as flap failure and reoperations) should be created to better assess the role of SMM depletion as a risk factor.

A cumulative variable including all the wound-related complications was created. It was added to the Materials and Methods sections. A table with univariate and multivariate analysis of factors associated with all the wound-related complications was added as a supplementary material. 

- As declared, the median time from CT image acquisition to total laryngectomy varied up to three months before surgery, potentially not being representative of the patients’ status at the time of surgery: this could have jeopardized the role of SMM depletion as a risk factor. A more limited inclusion criterion could be used.

Thank you for the comment. In the revised manuscript, the inclusion criteria have been modified to only include patients who underwent total laryngectomy for cancer. By excluding the group who did not have cancer (“dysfunctional larynx”), the median time interval between CT and surgery was of 37 days and the mean time interval was 60 days.

- A figure with an example of muscular segmentation and definition of skeletal muscle area (SMA), SMRA and intermuscular adipose tissue (IMAT) should be added for the sake of comprehension

A figure was added to the revised manuscript.

Round 2

Reviewer 1 Report

My questions and concerns have been adequately addressed.